# Strengthening Effect of Short Carbon Fiber Content and Length on Mechanical Properties of Extrusion-Based Printed Alumina Ceramics

**DOI:** 10.3390/ma15093080

**Published:** 2022-04-24

**Authors:** Haihua Wang, Jian Wu, Hai Zheng, Mingliang Tang, Xiaodong Shen

**Affiliations:** 1College of Materials Science and Engineering, Nanjing Tech University, Nanjing 211816, China; 201961103074@njtech.edu.cn (H.W.); 202061103110@njtech.edu.cn (J.W.); 2Donghai Institute of Advanced Silicon Based Materials, Nanjing Tech University, Nanjing 222300, China; 15996130575@163.com

**Keywords:** additive manufacturing, microstructure, mechanical properties, toughness

## Abstract

Extrusion-based ceramic printing is fast and convenient, but the green body strength is too low, and the application prospect is not high. An extrusion-based printing method of alumina ceramics toughened by short carbon fiber is reported in this paper. The bending strength and fracture toughness of 3D-printed alumina ceramics were improved by adding short carbon fiber. The toughening effects of four carbon fiber lengths (100 μm, 300 μm, 700 μm, and 1000 μm) and six carbon fiber contents (1, 2, 3, 4, 5, and 6 wt%) on ceramics were compared. The experimental results show that when the length of carbon fiber is 700 μm, and carbon fiber is 5 wt%, the toughening effect of fiber is the best, and the uniform distribution of fiber is an effective toughening method. Its bending strength reaches 33.426 ± 1.027 MPa, and its fracture toughness reaches 4.53 ± 0.46 MPa·m^1/2^. Compared with extrusion-based printed alumina ceramics without fiber, the bending strength and fracture toughness increase by 55.38% and 47.56%, respectively.

## 1. Introduction

Conventional ceramic-shaping methods, such as dry pressing, isostatic pressing, slip casting, tape casting, and injection molding [1,2,3,4,5], have limitations, and usually cannot be used for the fabrication of parts with complex shapes (internal holes, sharp corners, etc.) and parts requiring high accuracy. These processes are also rather time-consuming and have high costs because they require the fabrication of a mold and machining process (cutting, grinding, etc.) after sintering [6]. Fused used deposition modeling (FDM) [7], selective laser sintering (SLS) [8], ceramic jet printing (CJP) [9], stereolithography (SLA) [10], direct ink writing (DIW) [11], and extrusion free forming (EFF) [12] are some of the common methods that are used for additive manufacturing of ceramic and ceramic composite materials. Solid free forming can be defined as the creation of a shape by point, line, or planar addition of material without confining surfaces other than a base. The methods can, thus, be classified dimensionally, and extrusion free forming is in the group of linear methods [13]. Presenting a host of opportunities, 3D-printed scaffolds made out of polymers, ceramics, or composites are being widely investigated as candidates for dental tissue engineering [14,15,16]. Mousavi Nejad et al. used 3D printing technology to prepare two scaffold materials with different structures, and the results show that the two scaffolds have great potential for promoting hDPSC adhesion and odontogenic differentiation [17]. The application of 3D printing technology in ceramics has broad prospects, but the brittleness of ceramics always restricts the performance.

Aluminum oxide or alumina (Al_2_O_3_) is one of the most popular ceramic materials in additive manufacturing because of its excellent high-temperature resistance, compressive strength, hardness, wear resistance, and chemical stability [18,19]. In addition, ceramic alumina is suitable for a wide range of applications such as dielectric materials, substrates, and packaging for electronics, filters, optical components, implants, etc. [20]. The brittle behavior of ceramics and the underlying causes have been discussed extensively in the literature [21,22,23]. Ceramics do not undergo extensive plastic deformation before fracture, which is reflected in the fact that fracture work is a measure of toughness. In fact, the fracture work of most ceramics is about 10 J·m^−2^ [21]. Therefore, researchers used reinforcing agents such as single-wall carbon nanotube (SWCNT) and reduced graphene oxide (RGO) [24], MXene [25], alumina (Al_2_O_3_) whisker [26], carbon nanofibers (CNFs), and glass fiber to improve the toughness of alumina. Various toughening methods have made progress in experiments so that the mechanical properties of ceramics have been improved.

Carbon fiber can be used to improve the material properties of alumina ceramics. As regards 3D-printed alumina ceramics, we can also learn from this method to strengthen the toughness of ceramics. Extrusion-based ceramic printing has a fast forming speed and high solid content. It is a practical 3D printing ceramic technology. However, the extrusion bonding of slurry will reduce the performance of the final sintered body, which is far lower than the bending performance of traditional dry pressing. In this paper, alumina ceramics are printed by using rapid prototyping and extrusion-based printing methods, and carbon fiber is added to toughen the fracture toughness of ceramics. The effects of carbon fiber content, carbon fiber length, and mixed carbon fiber composition on the properties of ceramics are studied. This paper describes some results of an experimental investigation into the conditions required to achieve improvement in the mechanical properties of alumina ceramics by the incorporation of short carbon fibers. 

## 2. Experimental Procedure

### 2.1. Powder Mixture Preparation

First, alumina powder (LDW, D50 = 360 nm, BET = 6.07 m^2^/g, Foshan City Lidewang Trading Company Ltd., Foshan, China) was calcined at 900 °C in a muffle furnace (SXL-1216, Shanghai Jing Hong Laboratory Instrument Co., Ltd., Shanghai, China) for 1 h. Then, the powder was mixed with a certain amount of silica dioxide (SiO_2_, 0.3 wt%) and titanium dioxide (TiO_2_, 0.3 wt%) and milled with zirconia balls in deionized water (16.31 wt%) for 2 h. The suspension was dried in an oven at 60 °C for 2 h and finally screened out by a 325 mesh sieve. 

### 2.2. Preparation of the Ceramic Suspension

The premix solution consisted of poly (methacrylic acid) (PMAA-NH_4_, 0.8 wt%, dispersant), polyethylene glycol (2 wt%, plasticizer), water-soluble prepolymer UV (2 wt%, binder), glycerol (8 wt%, lubricant) and deionized water (87.2 wt%, solvent). Ammonia was used to adjust the pH to 9. The powder mixture was added to the premix with a mass fraction of 80 wt%, and the powder was then added in batches to prevent powder agglomeration. Then, the ceramic suspension was ball-milled with a zirconia ball for 2 h. It was stirred with a vacuum homogenizer (ZYMC-350VS, Shenzhen ZYE Science and Technology Co., Ltd., Shenzhen, China) for 30 min to remove bubbles in the ceramic suspension.

### 2.3. Preparation of the Carbon-Fiber-Reinforced Ceramic Suspension

Carbon fibers (All from Shanghai Liso Composite Material Technology Co., Ltd., Shanghai, China) with lengths of 100 μm, 300 μm, 700 μm, and 1000 μm were used in the experiment. The carbon fiber was soaked in absolute ethanol for 24 h and then washed with deionized water to remove the organic matter on the surface. The dried carbon fiber was soaked in concentrated nitric acid at 80 °C for 30 min, then washed with deionized water to become neutralized, and stored. Concentrated nitric acid can increase the surface activity of carbon fiber and strengthen the bonding between fiber and matrix.(1)Carbon fiber (size 300 μm) was added to the ceramic suspension with mass fractions of 1, 2, 3, 4, 5, and 6 wt%. The fiber and powder were stirred evenly under vacuum. The printed green bodies were sintered in an Ar atmosphere at 1100 °C, 1200 °C, 1300 °C, 1400 °C, and 1500 °C, respectively. The effects of carbon fiber content and sintering temperature on the compactness and strength of ceramics were studied.(2)The effect of composite carbon fiber on ceramic strength was studied. The added components of fiber are shown in Table 1. The mass fraction of the total fiber component accounted for 5 wt% of ceramic suspension, and the sintering temperature was 1500 °C. By comparing the effects of different fiber composite components on ceramic properties, the most appropriate carbon fiber composite ratio was obtained.

### 2.4. Fabrication of the Ceramic Parts through an Extrusion-Based 3D Printer

The entire Potter Artist-300 printer (Yueyang Dianfeng Electronic Technology Co., Ltd., Yueyang, China) includes a conveyor, an extrusion device, and a three-dimensional motion device. The conveying device includes a mud storage tank, a connecting pipe, and an air compressor, which is responsible for transmitting the prepared slurry to the machine head by air pressure. The extrusion device comprises a single screw, a cooling fan, and a nozzle, which presses the slurry evenly and continuously from the nozzle through the extrusion thrust of the screw. The three-dimensional motion device realizes the stable printing path through the program of the control panel, accurately prints the required structure of each layer, and controls the switch of the extrusion device. The printer device is shown in Figure 1.

The model was established by Magics v20.03_x64 software, and the appropriate printing parameters were set and imported into the machine for printing. Cylindrical blanks were printed with a diameter of 30 mm and a height of 10 mm. Keeping the UV light source in the printing process can make the photosensitive resin polymerize the alumina matrix better, so as to improve the compactness of ceramics.

Figure 2 shows the green shape of 3D-printed alumina ceramics. In the process of 3D printing, the printing rate, layer height, and nozzle diameter all affect the green forming. After many experiments, the printing parameters were adjusted to obtain the most suitable ceramic green forming conditions. In this experiment, the printing rate was 20 mm/s, the layer height was 1 mm, and the nozzle diameter was 1.2 mm.

### 2.5. Post-Processing of the Printed Green Body

The printed ceramic body was dried at 40 °C for 2 h and then gently separated from the template with a knife. The green body after demolding was dried at 50 °C for 24 h and then dried at 120 °C for 12 h, which is to prevent cracks and excessive residual water from drying too fast and damaging the sintering densification.

In a vacuum furnace (SK-G05163, Tianjin Zhonghuan Electric Furnace Co., Ltd., Tianjin, China), the first step was to heat the green body to 300 °C at a heating rate of 1 °C/min and hold it for 1 h; the second step was to heat the green body to 500 °C at a heating rate of 2 °C/min and hold it for 1 h; the third step was to heat the green body to 700 °C at a heating rate of 2 °C/min and hold it for 1 h; the fourth step was to heat the green body to 800 °C at a heating rate of 1 °C/min and hold it for 2 h. The sample was cooled to room temperature, and the degreasing curve is shown in Figure 3. The organic additives in the ceramic body were completely removed in the whole degreasing process. 

The degreased samples were sintered in a tubular furnace in an Ar atmosphere, the green body was heated to 1500 °C at a heating rate of 3 °C/min, held for 2 h, and then cooled to room temperature at the rate of 3 °C/min to obtain the sintered sample. 

### 2.6. Characterization

The relative density of the sintered samples D was measured by the Archimedes drainage method, as shown in Equation (1). The theoretical density of alumina ceramics ρ is 3.9 × 10^3^ kg/m^3^. The calculation formula is as follows:(1)$$\begin{array}{c} d=\frac{G_{1}\times d_{0}}{G_{2}-G_{3}}\\ D=\frac{d}{\rho } \end{array}$$
where *G*_1_ is the mass of the dried sample (kg); *G*_2_ is the mass of the sample saturated with water in air (kg); *G*_3_ is the mass of the sample saturated with water in water (kg); *d*_0_ is the density of water at the test temperature (kg/m^3^); *d* is the bulk density of the ceramic sample (kg/m^3^).

The bending strength *σ_f_* (MPa), shown in Equation (2), and fracture toughness K_IC_, shown in Equation (3), were measured on a universal material testing machine (CTM2000, Xie Qiang Instrument Manufacturing (Shanghai) Co., Ltd., Shanghai, China). The international standard was the GBT 6569-2006-Test method for flexural strength of fine ceramics. The sample was cut and ground to standard size. The bending strength was measured by the three-point bending method; the sample size was 35 mm × 4 mm × 3 mm, the span was 30 mm, and the loading rate was 0.5 mm/min. The calculation formula is as follows:(2)$$\sigma_{f}=\frac{3FL}{2bd^{2}}$$
where *L* is the span, taking 30 mm; *F* is the maximum load (N); *b* is the specimen width (m); *d* is the specimen height (m).

The most common notch beam method was used to determine the fracture toughness K_IC_ (MPa·m^1/2^), the aspect ratio was 2:1, and the span height ratio was 4:1. The sample size was 30 mm × 4 mm × 2 mm, the notch width was 0.25 mm, the notch depth was 1.7 mm, the span was 16 mm, and the loading rate was 0.05 mm/min. The test standard was single-edge notched bending (SENB). The calculation formula is as follows:(3)$$K_{IC}=Y\frac{3PL}{2bW^{2}}\sqrt{a}$$
where *L* is the span, taking 16 mm; *P* is the applied load (N); *W* is the sample height (m); *b* is the sample width (m); *a* is the crack length (m); *Y* is the dimensionless coefficient, in this experiment, *Y* = 1.93 − 3.07 (*a/W*) + 14.53 (*a/W*)^2^ − 25.07 (*a/W*)^3^ + 25.8 (*a/W*)^4^.

Each group of sintered bodies was tested 5 times, respectively, and the average value was taken as the total result. The microstructure of the sintered body was characterized by a scanning electron microscope (Apreo 2, Thermo Fisher Technology Co., Ltd., Bleiswijk, The Netherlands). The ceramic samples were crushed into small pieces, and the structure was observed by SEM after gold plating.

## 3. Results and Discussion

### 3.1. Effects of Sintering Temperature and Fiber Content on Properties of Alumina Ceramics

Figure 4 shows the bending resistance of alumina ceramics at different sintering temperatures. At low temperatures, the sintering of alumina ceramics was not dense enough, and a large number of pores appeared in the sintered body, resulting in a significant decrease in bending strength. The existence of pores also increases the expansion and extension of microcracks, separating the joint surface between carbon fiber and alumina matrix and reducing the fracture toughness of ceramics. Therefore, 1500 °C is the most suitable sintering temperature for alumina ceramic atmosphere sintering.

As depicted in Figure 4, when the sintering temperature was 1100 °C, the increase in carbon fiber content had no obvious reinforcing effect on alumina ceramics. However, with the increase in sintering temperature, the compactness of composite ceramics increased. The combination of ceramic matrix and fiber was closer and closer, and a greater external load would be required when a fracture occurred, indicating that the bending strength of the composites increased with the increase in carbon fiber content.

As shown in Figure 4, when short carbon fiber was added to the alumina matrix and sintered at 1500 °C, the bending strength of sintered body increased with the increase in carbon fiber content. When the fiber content increased, the interfacial bonding between carbon fiber and alumina matrix was enhanced, which plays a significant role in fracture, making the flexural strength of sintered body stronger than that of pure alumina ceramics. The bending strength of 3D-printed samples without any enhancement was 21.5 MPa. The addition of 1 wt% carbon-fiber-reinforced material increased the bending strength of the sample by 8.84% to 23.4 MPa, and the addition of 5 wt% carbon-fiber-reinforced material increased the bending strength by 47.44% to 31.7 MPa. 

Jinxing sun et al. used carbon-fiber-reinforced, 3D-printed, layered ceramic materials and found that they can improve the flexural strength and fracture toughness of the composites [27]. Alumina ceramics without fiber showed brittle fractures. After adding fiber, the fracture toughness of ceramics increased significantly. With an increase in fiber content, the fracture toughness of ceramics increased, and the deflection and bridging between fibers hindered the upward propagation of cracks.

However, when the fiber content reached 6 wt%, the nozzle of 3D printing was blocked due to the high content of carbon fiber, and the blank could not be printed. Even increasing the mixing time did not improve this situation. Therefore, taken together, it was concluded that 5 wt% was the maximum addition of carbon fiber that meets the conditions of extrusion-based printing.

The length of carbon fiber is 1000 μm, and because the nozzle diameter was close to the length of carbon fiber, it also led to blockage. Short carbon fibers are randomly distributed in ceramic slurry. When the fiber is too long, it cannot easily pass through the nozzle, resulting in the failure of the printing process. Therefore, we did not use a 1000 μm length of the fiber, to ensure the forming of a green body.

### 3.2. Effects of Different Fiber-Mixing Components on the Properties of Alumina Ceramic Sintered Body

Table 2 shows the average value and data deviation of bending strength and fracture toughness of each group of fiber 3D-printed ceramic composites. In terms of groups A, B, and C, 700 μm carbon fibers had the best reinforcement performance. This shows that the fiber with a large length can better combine with the interface of the alumina matrix to enhance the fracture performance of ceramics.

In the 3D printing process, the viscosity of the ceramic slurry is a very important printing parameter. Lower viscosity indicates that the dispersion is more uniform, which benefits the compactness of sintering. As shown in Figure 5, with the increase in carbon fiber content, the slurry viscosity increased, which indicates that alumina slurry had certain compatibility with carbon fiber.

Among the 10 groups, the bending strength and fracture toughness of group A were the highest, followed by groups B and F. It can be seen from Table 2 that the addition of fiber correspondingly increased the viscosity of the slurry and reduced the relative density. This is because the fiber destroys the dense structure of the ceramic and increases the pores, which reduces the compactness of the whole ceramic. Generally speaking, the decrease in compactness leads to a decrease in flexural strength of ceramic sintered bodies, but the addition of fiber also increases the toughness of ceramics. The balance between the two determined the final properties of composite ceramics.

Compared with the samples without fiber in group O, the addition of carbon fiber significantly improved the bending properties and toughness of 3D-printed alumina ceramics, and the decrease in ceramic compactness is not in an acceptable range. Among them, the toughening effect of all 700 μm carbon fibers was the best, the bending performance of pure alumina ceramics without fibers was the worst, and the performance of mixed fibers was between the two. Due to the mutual polymerization between long and short fibers, the viscosity of the slurry was reduced, but this way of destroying the uniformity of the slurry will lead to a decrease in strength, which was not as good as that of a single fiber on the whole. This shows that the fiber mixing between different lengths still needs to improve in experiments, and the uniformity of its dispersion should be increased.

### 3.3. SEM Image Analysis of Alumina-Sintered Body

Figure 6 shows the dispersion and fracture morphology of group A samples. Fiber-reinforced composites mainly include matrix failure, fiber–matrix interface debonding, fiber failure, fiber pullout, and delamination. When the crack propagates and expands in the matrix, this process may lead to fiber failure or fiber–matrix interface debonding. The debonding of the fiber–matrix interface also leads to fiber pullout. In our study, the layer of the 3D printing process was invisible or undetectable on the fracture surface. There was also no obvious delamination or any crack propagation between 3D printing layers. The fracture surface was composed of fiber pullout and fiber fracture, as shown in Figure 6a. As can be seen from Figure 6b, the carbon fiber became rough after surface treatment, and the adhesion with the alumina ceramic matrix was enhanced, resulting in the fracture of some fibers. This fiber fracture enhanced the fracture toughness and bending strength of ceramics.

The length of carbon fiber in group B was 300 μm. When the mass fraction was the same, the fiber-breaking energy provided by group B was not as good as that of group A, while the performance of group C was the worst. The SEM of group B samples is shown in Figure 7. It can be seen from Figure 7a that the carbon fibers were evenly distributed with fiber pullout and fiber fracture. From Figure 7b, it is revealed that the bonding between carbon fiber and alumina matrix was loose, which is conducive to fiber pulling out and reducing the interfacial bonding energy, resulting in a decrease in the overall strength of the composite. Therefore, it can be concluded that 700 μm was the most reinforcing among the three lengths of carbon fibers, the composite structure was compact, and the interface between the fiber and the matrix was well bonded.

From group D to group I, the mixed reinforcement effect of carbon fibers with different lengths was studied. It can be seen from Table 2 that the mixed carbon fibers are effective. The bending strength and fracture toughness of the samples were generally stronger than group C and weaker than group A, of which group F was the most obvious. Figure 8 shows the SEM of group F samples. In Figure 8b, it can be seen that the mixed fibers were arranged disorderly, and the fibers of different lengths entangled with each other. When the fiber is pulled out, some shorter fibers are brought out by the long fibers, leaving holes, which makes the strength of ceramics decline rapidly when they are broken and cannot maintain a certain stability. Although the performance of group F was better than that of group C, the disordered arrangement of group F made the fibers unable to play the role of cross-distribution and combined reinforcement of ceramics. In Figure 8a, it can be seen that due to the mixed distribution of fibers, some fibers were extruded, and their combination with the alumina matrix was too close, which led to partial corrosion during degreasing and sintering on the fiber surface. This was not observed in groups A and B. The carbon fibers in group F entangled with each other, further compressing the interface of the composite material, so that the surface corroded, resulting in a decline in the overall fracture performance. This shows that the fiber-mixing mode of group F needs to be improved to facilitate the dispersion of fibers. In 3D printing ceramics, compared with a single fiber, the mixed fiber is still not enough to obtain a good reinforcement effect, and there are many problems to be studied. We can perhaps use the double nozzle method to print the continuous fiber reinforcement, but the winding of the printing path is a considerable problem. We can also use the oscillation method to mix the fibers to disperse evenly as much as possible, but it is difficult to ensure the porosity of ceramic materials without vacuum. Although the relevant experimental schemes can still be explored, we still need to perform more tests and solve various problems.

## 4. Conclusions

The ceramic paste with a viscosity suitable for extrusion-based printing was successfully prepared by mixing deionized water and aluminum oxide powder, adding additives and carbon fiber. When the carbon fiber content was 6 wt%, or the carbon fiber length was 1000 μm, the slurry formed a blockage at the nozzle, resulting in printing failure. The properties of ceramic sintered bodies with five carbon contents (1, 2, 3, 4, and 5 wt%) and three carbon fiber lengths (100 μm, 300 μm, and 700 μm) were tested, and the toughening effects of mixed fibers with different fiber lengths were also tested. Mixed fibers could not be well used in reinforced 3D-printed ceramics, while single fibers could better enhance the fracture performance of ceramics, and the fibers were evenly distributed and well combined with the matrix.

Short carbon-fiber-reinforced alumina ceramic composites were prepared using a 3D printer. SEM image analysis showed that 700 μm carbon fiber was evenly dispersed, and there were both fiber fracture and fiber pullout on the fracture surface; fiber pullout is the main fracture mechanism. When the total content of carbon fiber was 5 wt%, the toughening effect of 700 μm carbon fiber was the best, the bending strength reached 33.426 ± 1.027 MPa, and the fracture toughness reached 4.53 ± 0.46 MPa·m^1/2^. Compared with 3D-printed alumina ceramics without fiber, the bending strength increased by 55.38%, and the fracture toughness increased by 47.56%.

## Figures and Tables

**Figure 1 materials-15-03080-f001:**
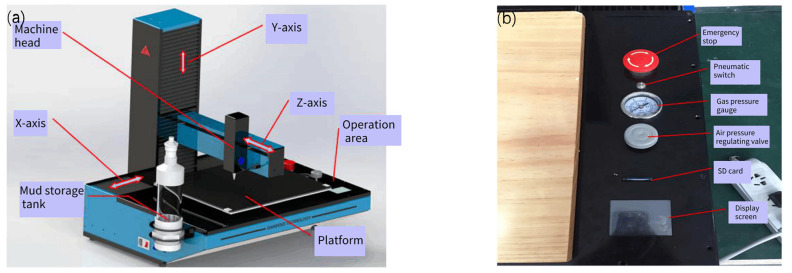
Equipment diagram of Potter Artist-300 printer: (**a**) each component; (**b**) operation interface modules.

**Figure 2 materials-15-03080-f002:**
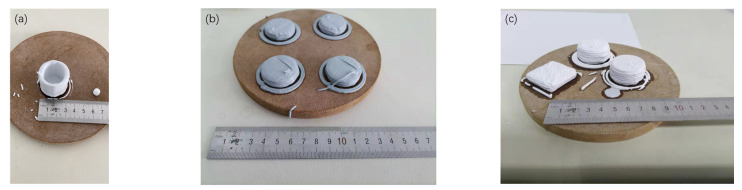
Extrusion-based mold printing of alumina ceramics: (**a**) 30 layers of high-hollow pure alumina body; (**b**) alumina green body with carbon fiber added; (**c**) solid pure alumina green bodies with different shapes.

**Figure 3 materials-15-03080-f003:**
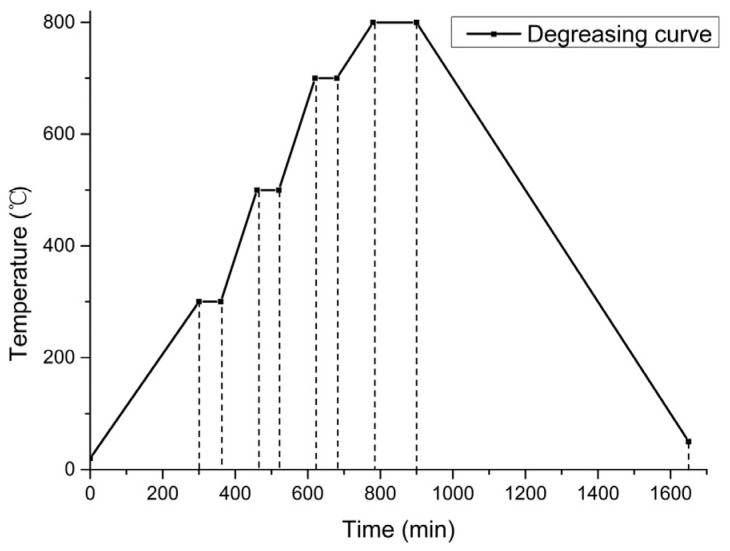
Degreasing curve of sample in vacuum furnace.

**Figure 4 materials-15-03080-f004:**
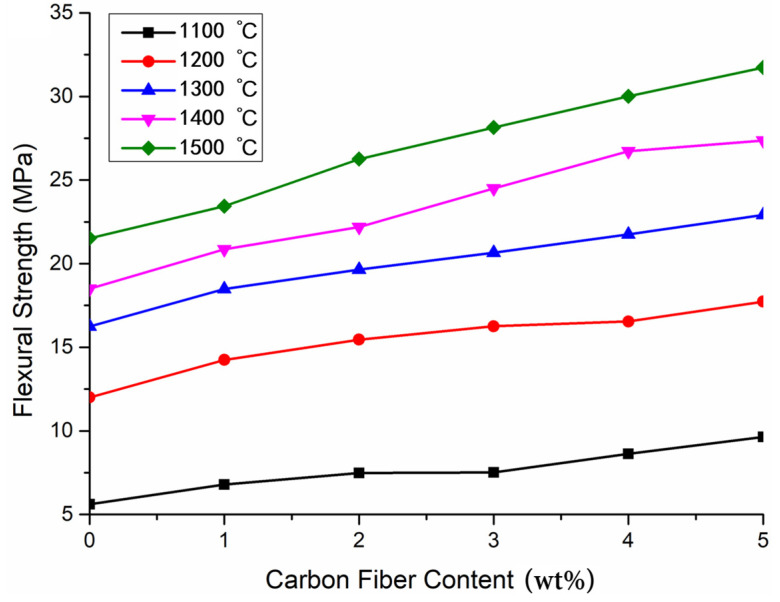
Bending strength of ceramic-fiber-reinforced materials at different sintering temperatures.

**Figure 5 materials-15-03080-f005:**
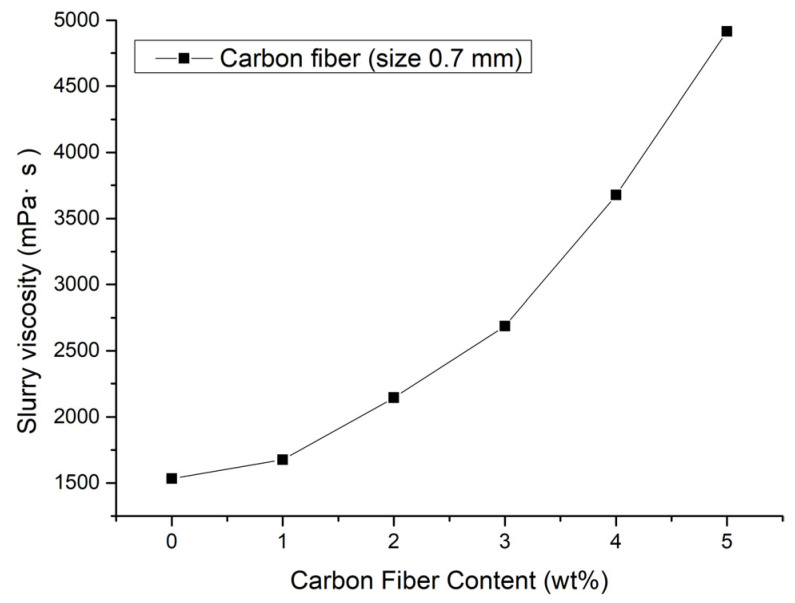
Variation of viscosity with carbon fiber content.

**Figure 6 materials-15-03080-f006:**
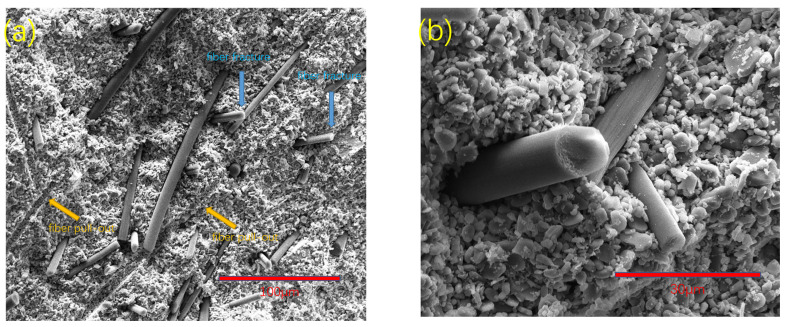
The 3D-printed SEM images of fracture of group A samples of alumina composite ceramics: (**a**) with 500× magnification and a scale of 100 μm, fiber pullout and fiber fracture were observed; (**b**) with 2000× magnification and a scale of 30 μm, a clear image of fiber fracture can be seen.

**Figure 7 materials-15-03080-f007:**
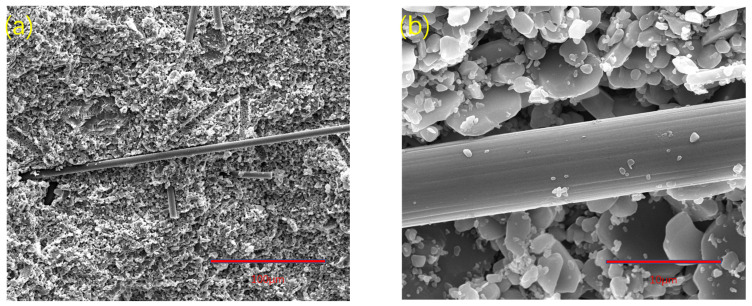
The 3D-printed SEM images of fracture of group B samples of alumina composite ceramics: (**a**) with 500× magnification and a scale of 100 μm, the matrix and fiber were evenly distributed; (**b**) with 5000× magnification and a scale of 10 μm, the interface between matrix and fiber was well bonded.

**Figure 8 materials-15-03080-f008:**
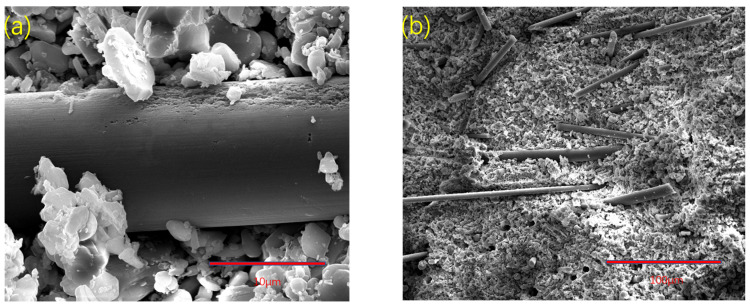
The 3D-printed SEM images of fracture of group F samples of alumina composite ceramics: (**a**) with 5000× magnification and a scale of 10 μm, the fiber surface partially corroded; (**b**) with 500× magnification and a scale of 100 μm, the fiber arrangement was chaotic, and some were densely stacked.

**Table 1 materials-15-03080-t001:** Short carbon fibers of different lengths were mixed. The proportion of total fiber was 5 wt%, and group O is 3D-printed ceramic samples without fiber.

Length	Mass Fraction/wt%									
Sample Reference	A	B	C	D	E	F	G	H	I	O
0.7 mm	100	0	0	5	15	80	10	30	60	0
0.3 mm	0	100	0	15	80	5	30	60	10	0
0.1 mm	0	0	100	80	5	15	60	10	30	0

**Table 2 materials-15-03080-t002:** At a sintering temperature of 1500 °C, the mechanical properties of 3D-printed alumina ceramics reinforced with different components of short carbon fiber were compared, and group O is 3D-printed ceramic samples without fiber. The data in the table are the average results.

Sample Reference	Slurry Viscosity (mPa·s)	Relative Density	Bending Strength (MPa)	Fracture Toughness (MPa·m^1/2^)
A	4917.21	87.20%	33.426 ± 1.027	4.53 ± 0.46
B	3873.79	88.15%	31.684 ± 1.164	4.36 ± 0.34
C	3456.42	87.16%	28.835 ± 0.756	3.54 ± 0.27
D	2115.62	92.46%	29.734 ± 0.945	3.73 ± 0.48
E	2713.41	90.24%	29.242 ± 1.725	3.84 ± 0.19
F	3170.96	90.16%	32.610 ± 1.049	4.32 ± 0.43
G	1887.99	93.24%	29.434 ± 1.226	3.82 ± 0.37
H	2302.22	91.42%	29.472 ± 0.452	4.02 ± 0.64
I	2890.65	91.61%	30.446 ± 1.049	4.15 ± 0.29
O	1533.44	96.38%	21.512 ± 0.629	3.07 ± 0.28

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
