# Peer review of "Strengthening Effect of Short Carbon Fiber Content and Length on Mechanical Properties of Extrusion-Based Printed Alumina Ceramics"

_materials, 2022, doi:10.3390/ma15093080_

Round 1

Reviewer 1 Report

Please kindly consider the attached report

Author Response

Dear Reviewers,

Thank you very much for your valuable comments. This is very helpful for the improvement of our article.

  1. Abstract: it has been well written and the authors have briefly presented the results

The abstract has been revised to better suit the research content.

  1. Introduction: in the first paragraph of introduction (line 28), please kindly consider adding the following reference for mentioning FDM/FFF beside ref 7: https://doi.org/10.1016/j.jmapro.2022.02.042

Thank you for your recommendation. The literature has been added.

  1. It seems that Figure 1 is a two sub-figure. Consider adding a and b as well as the caption to define the explanation of each figure.

   The drawing has been modified.

Thank you for your patient guidance. We have revised the paper according to your opinions.

Wish you a happy life.

Kind regards

Reviewer 2 Report

The manuscript is interesting, but it has many issues. There are still some pre-submission parts that should be removed before the submission. I listed all issues below:

  1. In the abstract part, the authors should highlight the most important outcomes. In the present form, the abstract looks like a part of the methodology and results. Please look at the guide for the authors in the journal's part - there are all informations about the abstract form. 
  2. Please provide the whole experimental description parts where you will provide all data about the used apparatus in your research. 
  3. The text should be not colored (lines 175-179). 
  4. Please provide the statistical analysis of the data shown in table 2
  5. How did the authors measured the density of samples? Please provide the methodology description. 
  6. How many samples were tested for each combination? 
  7. Please provide the standard for all strength tests which you made. 
  8. Your conclusion is formed in that way that you failed, and there are not any advantages in your work. Please rewrite it and find some novel outcomes which could improve the present state of the art. 

I think this manuscript has the potential to be published but it has to be significantly improved. 

Author Response

Dear Reviewers,

Thank you very much for your valuable comments. This is very helpful for the improvement of our article.

  1. In the abstract part, the authors should highlight the most important outcomes. In the present form, the abstract looks like a part of the methodology and results. Please look at the guide for the authors in the journal's part - there are all informations about the abstract form.

The abstract has been revised to better suit the research content.

  1. Please provide the whole experimental description parts where you will provide all data about the used apparatus in your research.

Thank you for your recommendation. Relevant instruments have been added.

  1. The text should be not colored (lines 175-179). 

    OK, it has been modified.

  1. Please provide the statistical analysis of the data shown in table 2.

The average value of 5 samples in each group is described in Table 2, and the data deviation is added. The purpose of using tables is to show the accuracy of data more clearly.

  1. How did the authors measured the density of samples? Please provide the methodology description. 

A formula has been added to calculate the relative density of ceramics.

  1. How many samples were tested for each combination? 

Basically, each group tested 5 samples and took the average value list.

  1. Please provide the standard for all strength tests which you made. 

“GBT 6569-2006 Test method for flexural strength of fine ceramics” is used to test the bending strength.

The fracture toughness of ceramics was measured by single edge notched bending (SENB).

  1. Your conclusion is formed in that way that you failed, and there are not any advantages in your work. Please rewrite it and find some novel outcomes which could improve the present state of the art. 

The results have been re demonstrated to highlight the progress of this experiment. Please review whether it is appropriate.

Thank you for your patient guidance. We have revised the paper according to your opinions.

Wish you a happy life.

Kind regards

Reviewer 3 Report

Dear Authors,

Congratulations on your work, which is focused on a very interesting subject. As any other paper in this phase, there are some amendments to do, whose can improve the overall quality of your paper. Thus, I'm providing below some comments and suggestions, trying to collaborate by this way in improving your paper:

  1. 1. The Abstract doesn't clearly state the literature gap found, as well as the main motivation to develop this work. Thus, please clearly state the gap found in the literature in the Abstract, Introduction and Conclusions. The mains goals are also not clear in the Abstract.
  2.  The novelty brought by your work is also not properly pointed out. Thus, please state clearly the novelty that your paper represents for the scientific community, stating as well if your contribution is exclusively scientific or if there was some practical motivation behind the development of your work. Any industrial application based on this work should also be pointed out.
  3. Please avoid large groups of references allocated to just one idea (example: [1-5].
  4. The Introduction is poor because no direct speech is used. Please refer previous works carried out in similar way than this one, and refer the main goals and main achievements obtained regarding each one.
  5. Please point out the brand name and model of the equipments used in the experimental.
  6. Please use always a blank space between values and units, as recommended by standards.
  7. Figure 1 is too basic, i.e., unnecessary. Please remove it.
  8. In Figure 3 please use dashed vertical lines to help reader to easily locate the time for each step.
  9.  When describing the variables contained in each formula, please point out the units using the International Units System.
  10. The head of some tables don't help the reader to easily understand what is seeing.
  11. The scale in Figure 5 is not easy to understand. Please improve.
  12. I don't know why some text is in blue colour. Could you explain?
  13. No discussion is provided, which is a severe lack, don't helping the reader to understand the positioning of your results.
  14. As much as possible, please refer the number of experiments performed and provide always the standard deviation in the calculations.
  15. The Conclusions don't sound. Please reinforce and highlight the main novelty of your work.

Hope these comments help you in improving your paper.

Kind regards.

Author Response

Dear Reviewers,

Thank you very much for your valuable comments. This is very helpful for the improvement of our article.

  1. The Abstract doesn't clearly state the literature gap found, as well as the main motivation to develop this work. Thus, please clearly state the gap found in the literature in the Abstract, Introduction and Conclusions. The mains goals are also not clear in the Abstract.

The abstract has been revised to better suit the research content. Thank you for your guidance.

  1. The novelty brought by your work is also not properly pointed out. Thus, please state clearly the novelty that your paper represents for the scientific community, stating as well if your contribution is exclusively scientific or if there was some practical motivation behind the development of your work. Any industrial application based on this work should also be pointed out.

The introduction is revised to highlight the contribution of this paper.

  1. Please avoid large groups of references allocated to just one idea (example: [1-5].

Sorry, this is the summary of the previous methods. If you take it apart for reference, the article will be too dense. In the following quotation, we have tried to disassemble it.

  1. The Introduction is poor because no direct speech is used. Please refer previous works carried out in similar way than this one, and refer the main goals and main achievements obtained regarding each one.

OK, the introduction has been revised.

  1. Please point out the brand name and model of the equipments used in the experimental.

Device parameters added.

  1. Please use always a blank space between values and units, as recommended by standards.

Related errors have been modified.

  1. Figure 1 is too basic, i.e., unnecessary. Please remove it.

Sorry, the devices related to various 3D methods are different. I think it's worth a brief introduction.

  1. In Figure 3 please use dashed vertical lines to help reader to easily locate the time for each step.

OK, the dotted line in the figure has been added.

  1. When describing the variables contained in each formula, please point out the units using the International Units System.

OK, it has been modified.

  1. The head of some tables don't help the reader to easily understand what is seeing.

The header description of each chart has been modified.

  1. The scale in Figure 5 is not easy to understand. Please improve.

Add the instruction in Figure 5 and modify the unit value in the text to match the figure.

  1. I don't know why some text is in blue colour. Could you explain?

This is the modified part. Sorry, it has been changed back to black.

  1. No discussion is provided, which is a severe lack, don't helping the reader to understand the positioning of your results.

A discussion section has been added to better illustrate the results.

  1. As much as possible, please refer the number of experiments performed and provide always the standard deviation in the calculations.

In the experiment, five groups of tests were carried out in each group, and the average number was obtained. Data deviation has been added.

  1. The Conclusions don't sound. Please reinforce and highlight the main novelty of your work.

The conclusion is revised to highlight the research focus and progress of this experiment.

Thank you for your patient guidance. We have revised the paper according to your opinions.

Wish you a happy life.

Kind regards

Reviewer 4 Report

The article describes the improvement of mechanical properties such as strength and toughness of 3D-printed alumina ceramics. The minor revision is required.

  • The strength was almost 1/10 of commercial alumina ceramics. Why is that? Refer the strength of 3D printed alumina ceramics and so the purpose of this article needs to be strengthened.
  • How much is the toughness of alumina without fiber addition? Present it and clarify the toughness is improved by fiber addition.
  • How many data of Fig. 4 and Table 2.? Add the standard deviation. At leat, 3 samples are needed to measure the bending strength. 
  • Revise  0.1, 0.3, 0.7 mm to 100, 300, 700 micrometers because the unit of scale bar of SEM pictures is micrometer.  

Author Response

Dear Reviewers,

Thank you very much for your valuable comments. This is very helpful for the improvement of our article.

  • The strength was almost 1/10 of commercial alumina ceramics. Why is that? Refer the strength of 3D printed alumina ceramics and so the purpose of this article needs to be strengthened.

Extrusion printing ceramics are printed layer by layer with air compressed slurry. The density of printing is not as high as that of dry pressing, and the final strength is very low. Among various 3D printing technologies, extrusion printing is fast and simple, but the strength is not high, so we study the effect of carbon fiber reinforcement.

  • How much is the toughness of alumina without fiber addition? Present it and clarify the toughness is improved by fiber addition.

Thank you for your reminder. The data without carbon fiber has been added for comparison.

  • How many data of Fig. 4 and Table 2.? Add the standard deviation. At leat, 3 samples are needed to measure the bending strength. 

In this experiment, we basically made five identical samples for each group. The data in the chart is the average. Now we have added data deviation to increase the reliability.

  • Revise 0.1, 0.3, 0.7 mm to 100, 300, 700 micrometers because the unit of scale bar of SEM pictures is micrometer.  

OK, thanks for your reminder. It has been revised.

Thank you for your patient guidance. We have revised the paper according to your opinions.

Wish you a happy life.

Kind regards

Reviewer 5 Report

The manuscript entitled “Strengthening Effect of Short Carbon Fiber Content and Length on Mechanical Properties of 3D Printed Alumina Ceramics” authored by Haihua Wang, Jian Wu, Hai Zheng, Mingliang Tang and Xiaodong Shen has been reviewed.

This paper investigates a 3D printing method of carbon fibre toughened alumina ceramics. Densification on sintering is improved by configuring powder, then the ratio is adjusted to obtain ceramic slurry with high solid content and plasticity, and the surface of carbon fiber is pre-treated to combine closely alumina matrix. The results show that at 1500℃ and 5 wt% carbon fiber content, the flexural strength and fracture toughness of the alumina ceramic sintered body enhanced significantly compared to plain alumina ceramics. The reinforcing effect of 0.7 mm carbon fiber is easy-to-accomplish, and outstanding without affecting the basic structure. The paper reports a novel and very interesting concept, which is worth publishable. Overall the manuscript is very well organized, but lacks flow of the English language. There are plenty of references but have problems with English grammar and some minor corrections described below before publication.

Q1. P1, L11-13, “Densification on sintering is improved by configuring powder, then the ratio is adjusted to obtain ceramic slurry with high solid content and plasticity, and the surface of carbon fibre is pre-treated to combine closely alumina matrix”.

Please correct the English grammar.

Q2. P1, L40-44, “Aluminum oxide or alumina (Al2O3), is one of ceramic materials favored in additive manufacturing, due to its excellent properties including high temperature resistance, compressive strength, hardness, wear resistance and chemical stability. In addition, ceramic alumina is suitable for a wide range of applications such as dielectric materials, substrate and package for electronics, filters, optical components, implants.”

Please correct the grammar of the sentence

Q3. P2, L46-51, “It occurs because ceramics do not undergo extensive plastic deformation prior to fracture and is reflected in the fact that the work of fracture, which is a measure of toughness, is for most ceramics of the order of 10 J·m-2[21]. Therefore, additive of reinforcement such as Single wall carbon nanotube (SWCNT) and reduced graphene oxide (RGO) [24], MXene [25], alumina (Al2O3) whisker [26], carbon nan-ofibers (CNFs), and glass fiber have been used to improve the toughness of alumina.”

Please correct the English grammar

Q4. P2, L59-61, “Then, the powder was mixed with a certain amount of silica dioxide (0.3 wt%) and titanium dioxide (0.3 60 wt%) and milled with zirconia balls in deionized water (16.31 wt%) for 2 h.”

Please change “silica dioxide” as “silicon dioxide (SiO2).

Q5. P2, L-63 and L71, the side headings of Experimental Procedure is written as

2.2. Preparation of the ceramic suspension and 2.3. Preparation of the ceramic suspension

Please change the side heading of section 2.3 as:  2.3. Preparation of the carbon fiber reinforced ceramic suspension, Or whatever the author’s choice.

Q6. P2, L 73-77, “The carbon fiber was soaked in absolute ethanol for 24 h to remove the surface organic matter, and then washed with deionized water and dried. Then soak it in concentrated nitric acid at 80℃ for 30 minutes to increase the surface activity of carbon fiber, then wash it with deionized water to neutral, and then dry and save it”.

Please correct the sentence structure. Please explain it the way you have done the task.

Q7. P2, L 78-79, “1)0.3mm carbon fiber was added to the ceramic suspension with mass fractions of 1, 2, 3, 4 and 5 wt%”.

Please change the sentence as: 1) Carbon fiber (size 0.3 mm) was added to the ceramic suspension with mass fractions of 1, 2, 3, 4 and 5 wt%

Q8. P3, L-114-115, “Dry the printed ceramic body at 40℃ for 2 h, and then gently separate the ceramic body from the template with a knife”.

Please correct the sentence structure

Q9. P4, L-115-117, “The green body after demoulding was dried at 50℃ for 24 h, and then dried at 120℃ for 12 h. This is to prevent cracks and too much residual water from drying too fast, which will affect the densification of sintering”.

Please rewrite the sentence to get a flow

Q10. P4, L-118-124, “In the vacuum furnace, the first step is to heat the green body to 300℃ at a rate of 1 ℃/min and held kept warm for 1 h, the second step is to heat the green body to 500℃ at the rate of 2 ℃/min and kept warm for 1 h, the third step is to heat the green body to 700℃ at the rate of 2 ℃/min and kept warm for 1 h, and the fourth step is to heat the green body to 800℃ at the rate of 1 ℃/min and kept warm for 2 h. The sample is cooled to room temperature, and the degreasing curve is shown in Figure 3. The organic additives in the ceramic body are completely removed in the whole degreasing process”.

Please change the clause “held kept warm for 1 h” as follows:

heat the green body to 300℃ at a heating rate of 1 ℃/min and hold it for 1 h,  and repeat the same throughout the paragraph.

Q11. P4, L-31-132, “The bulk density of the sintered samples was measured by Archimedes drainage method [equation (1)].

Please correct English grammar. Please modify the sentence as follows: The bulk density of the sintered samples was measured by Archimedes drainage method as shown in equation (1). Also, please define “d” in equation (1). Also the equation numbers (1), (2) and (3) on the right side are misaligned. Please align them straight towards the respective equations as well.

Q12. P4, L-137-138, “The bending strength [equation (2)] and fracture toughness [equation (3)] were measured on a universal material testing machine”.

Please indicate the make/model of the universal material testing machine used in this experiment for the characterization of the mechanical properties such as 3 point bending strength as well as fracture toughness. Also please indicate the ASTM standards referred in the respective measurement. Give references.

Q13. P4, L-137-138, Define “σf ” in the equation (2)

Q14. P4, L-153-154, “The microstructure of the sintered body was characterized by scanning electron microscope”.

Please indicate the description of the scanning electron microscope (SEM) used in this experiment. Briefly explain the sample preparation and description of the microscopy experimental conditions to take the images.

Q15. P5, L-173-174, “In Figure 4: Bending strength of ceramic fiber reinforced materials at different sintering temperatures”.

How many samples you studied for flexural strength measurements? Did you consider the errors? What is the standard deviation of the        flexural strength? Please provide the error bars on the diagram (Figure 4). Also we can see the graphs are broken at the maximum points on the right border of Figure 4. Please adjust the X-axis scale so that the plots will be located well inside the right border.

Q16. P5, L175-179, “When the sintering temperature is 1100℃, the increase of carbon fiber content has no obvious reinforcing effect on alumina ceramics. However, with the increase of sintering temperature, the connection between matrix and fiber becomes closer and closer, showing the law that the flexural strength of composites increases with the increase of carbon fiber content.”                                                                                         

This part of the discussion is not clear. Please elaborate the content meaningfully.

Q17. P6, L186-188, “3.2. Effects of fibers with different components on the properties of alumina ceramic sintered body. Table 2 shows the bending strength of 3D printed ceramic composites of carbon fibers with different components.

What do you mean by different components? Please clarify/correct the statement.

Q18. P6, L186-188, “the fiber with large length can better combine with the interface of alumina matrix to enhance the fracture performance of ceramics.”

Briefly explain why fiber with large length can enhance the fracture performance of ceramics? Give reference.

Q19. P6, L191-192, “In the 3D printing process, the viscosity of ceramic slurry is a very important printing parameter”.

Please provide a graph showing viscosity vs carbon fiber content. Did you use any deflocculates in the ceramic slurry? How was the consistency of the alumina slurry with carbon fiber?

Q20. P6, L196-197, “Table 2. The mechanical properties of 3D printed alumina ceramics reinforced with different components of short carbon fiber were compared”.

What was the sintering temperature of the samples studied for the mechanical property measurements? Please mention the sintering temperature on Table 2 title.

Q21. P6, L193-195, “Among the nine groups, the viscosity of group G is the lowest, and the bending strength and fracture toughness of group A are the highest, followed by group B and group F.”

Table 2. The mechanical properties of 3D printed alumina ceramics reinforced with different components of short carbon fiber were compared.

According to Table 2, Group G has the lowest viscosity but highest density and lower mechanical properties compared to Group A and B. How do you connect the Density, Bending strength and fracture toughness? Explain.

Q22. P6, L200-203, “Figure 5 shows the dispersion and fracture morphology of group A samples through SEM images. fiber reinforced composites mainly include matrix failure, fiber/matrix interface debonding, fiber failure, fiber pull-out and delamination.”

Please correct the sentence structure

Q23. P7-8, “SEM micrographs shown in Figure 5, Figure 6 and Figure 7 the magnification scale marked (in red colour) on the pictures are not readable. Please make them clear.

Q24. P7-8, L214-248, “The length of carbon fiber in group B is 0.3 mm. When the mass fraction is the same,… the fiber surface is partially corroded; (b) 500 × magnification, the fiber arrangement is chaotic, and some are densely stacked.”

This part of the manuscript is misaligned. Please align it.

Q25. P8, L242-245, “This shows that the fiber mixing mode of group F needs to be improved to facilitate the dispersion of fibers. In 3D printing ceramics, compared with a single fiber, the mixed fiber is still not enough to obtain good reinforcement effect, and there are many problems to be studied.”

How can we improve the fiber mixing mode of a mixed fiber reinforced ceramic slurry? What are the possible issues we need to address in the case of mixed fiber reinforced slurry and the plausible solutions?

Author Response

Dear Reviewers,

First of all, I'm sorry to have so many questions for you to review. Thank you for your careful guidance.

I have revised many of your questions one by one. I hope you can check my paper again. If you have any comments, please feel free to contact me.

Thank you again for your help.

Wish you a happy life.

Kind regards

Round 2

Reviewer 1 Report

Good luck

Author Response

Dear Reviewers,

Thank you for your guidance during this time.

Wish you a happy life.

Kind regards

Reviewer 2 Report

All corrections are properly made. Congratulations. 

Author Response

Dear Reviewers,

Thank you for your careful guidance. I also revised the paper a little bit. This is the result of cooperation.

Wish you a happy life.

Kind regards

Reviewer 3 Report

Thank you so much for addressing my comments and suggestions.

Author Response

Dear Reviewers,

It's very kind of you. I still have a lot of knowledge to learn. Thank you for your patient guidance during this time.

Wish you a happy life.

Kind regards

This manuscript is a resubmission of an earlier submission. The following is a list of the peer review reports and author responses from that submission.

Round 1

Reviewer 1 Report

Manuscript ID: MDPI Materials-1642506

Title: Enhancement Effect of Short Carbon Fiber on 3D Printed Alumina Ceramics

The work basically describes the additive manufacturing of carbon fibre reinforced alumina. The authors have studied, based on an experimental design, different short carbon fibres and have identifed some of the  prepared composition to have good mechanical properties and hence on.

The study is well planned; the processing parts: powder mixture preparation, carbon fibre introduction, 3D printing and, the characterization part are well explained. However, when it comes to the results and discussion section of the manuscript, it is losing its charm. However, the work can be considered as a short report (as a communication) or should be sent back requesting more experiments to be added so that to consider it as a good original article. In any case, if the Editor feels to considered it, there are some basic points which need to be addressed:

Title need to be little more specific : Simply telling the enhancement doesn’t go good. Enhancement of what by carbon fibres??

Abstract:

A 3D printing method of carbon fiber toughened alumina ceramics is reported in this paper. The sintering density The sintering density of ceramic is improved by configuring powder, then the ratio is…..

Sintering density sounds little weird. ‘Densification on sintering’ can be understood.Please consider to revise such statements.

  • and a pots-machining processing …typos need to be corrected
  • The brittle behaviour of ceramics and the underlying causes have been discussed extensively in the literature….Please add those reference literatures
  • Figure 1. Structure diagram of Potter Artist-300 printer …. It is a picture of the device with the parts labelled…not a structure diagram according to me
  • Figure 3. The bending strength of 3D printed alumina ceramics is enhanced by different sintering temperature and carbon fiber content:(a) Bending strength of ceramic fiber reinforced materials at different sintering temperatures; (b) Flexural strength of different carbon fiber content reinforced materials at 1500°C sintering temperature.
  • From what I know, The bending strength and the flexural strength are the same. Why did the authors represent them separately in two different figures. If that is the case: it is repetition of the data presentation and should be changed. The sample bending strength. of samples presented in figure 3(a) at 1500 ºC (green plot) is same as the bar chart shown in the figure 3(b)?
  • Figure 4. Fracture surface of short carbon fiber reinforced 3D printed alumina ceramic sample. (a)fiber pull-out; (b)fiber fracture.

 In the SEM images of the samples presented: please modify the figure with only the scale and cutting the black bar with the magnification shown. The magnification doen’t makes any sense.

Author Response

Thank you very much for your guidance. I have carefully revised the article with reference to your review.

Title need to be little more specific : Simply telling the enhancement doesn’t go good. Enhancement of what by carbon fibres??

The title was revised to better suit the article.“Strengthening Effect of Short Carbon Fiber on Mechanical Properties of 3D Printed Alumina Ceramics”.

Sintering density sounds little weird. ‘Densification on sintering’ can be understood.Please consider to revise such statements.
•    and a pots-machining processing …typos need to be corrected

Related errors have been corrected.

The brittle behaviour of ceramics and the underlying causes have been discussed extensively in the literature….Please add those reference literatures

Relevant references have been added.

Figure 1. Structure diagram of Potter Artist-300 printer …. It is a picture of the device with the parts labelled…not a structure diagram according to me

The descriptive error in Figure 1 has been corrected.

From what I know, The bending strength and the flexural strength are the same. Why did the authors represent them separately in two different figures. If that is the case: it is repetition of the data presentation and should be changed. The sample bending strength. of samples presented in figure 3(a) at 1500 ºC (green plot) is same as the bar chart shown in the figure 3(b)?

Thank you for your review. This is repetitive data entry. The duplicate chart has been deleted.

 In the SEM images of the samples presented: please modify the figure with only the scale and cutting the black bar with the magnification shown. The magnification doen’t makes any sense.

Figure(b) is not only an enlarged image, but also explains the importance of early surface treatment of carbon fiber.Relevant instructions have been added.

Finally, your review is very helpful for the improvement of the article. Thank you for your efforts.

Reviewer 2 Report

Please kindly consider the attached file

Author Response

Thank you very much for your review during this period, which is very helpful for the revision of our article.

1.A general comment goes to the presentation of the Figures. I request you take action

regarding the modification of the size and quality of both figures and curves.

The error caused by repetitive data has been corrected. I hope it can be as concise and clear as possible.

2.Also, please kindly consider adding equation numbers for each equation, as the way of

presentation is not suitable for a research article. In page 4, it is required to put the

equations with their equation number between the phrases.

The relevant equation format has been revised and explained.

3.Figure 4: the scale of the SEM images is not clear. Please revise.

SEM images have been uploaded and modified again.

4.The introduction is too short and there is not enough bibliographic studies. In the

introduction, in the first paragraph, it is required to briefly talk about the different

application of Additive Manufacturing.

References have been added in the introduction to enrich the content.

Finally, thank you very much for your hard guidance and wish you a happy life.

Reviewer 3 Report

Dear Authors,

First of all, congratulations on your work which is about a very interesting subject. As any other initial paper, there is room for improvement, thus, I'm providing you some suggestions and comments below, helping in improving your paper.

  1. Please proofread your paper carefully, avoiding mistakes such as: "...pots-machining...";
  2. In the first paragraph of the Introduction, there are a series of lacks of spaces betweem words. Please rectify this in all manuscript;
  3. It would be expected that Introduction includes some results of other works about the reinforcement of ceramics in Additive Manufacturing. Could you enrich the Introduction with some results of previous works, allowing for a better comparison with your results in your Discussion?
  4. Please use always a blank space between values and units. This is not coherent in your manuscript.
  5. In 2.1 is not clear the proportion of each component used in the mixing process. Could you clarify?
  6. In Table 1, please change the samples' reference to the 2nd line, and change also the name from "Number" to "Sample's Reference". Please do the same in Table 2;
  7. Please refer the real brand of "magics software" (line 98);
  8. In line 99, I thing there is something wrong: "...and the printer is imported for printing." Please correct.
  9. In 2.5 the text is hard-to-read. Please rewrite it with the help of an English Native Speaker;
  10. In 2.5 please include a diagram with the thermal cycle.
  11. Please format 2.6. All formulas need to be numbered;
  12. Please explain why you have choosen Y≈ 2.1639;
  13. Please include a Discussion in your paper, halping the reader to compare the results previously obtained in other similar works and yours;
  14. It would be expected a deeper discussion about the phenomena behind the results. Thus, please enrich your manuscript crossing experiences with other Researchers.

Best wishes for your work.

Kind regards.

Author Response

Dear Reviewers,

Thank you very much for your valuable comments. This is very helpful for the improvement of our article.

1.Please proofread your paper carefully, avoiding mistakes such as: "...pots-machining...";

The school team thesis has been revised. Thank you for your careful guidance.

2.In the first paragraph of the Introduction, there are a series of lacks of spaces betweem words. Please rectify this in all manuscript;

The error has been corrected.

3.It would be expected that Introduction includes some results of other works about the reinforcement of ceramics in Additive Manufacturing. Could you enrich the Introduction with some results of previous works, allowing for a better comparison with your results in your Discussion?

Add references to enrich the content and increase the depth of discussion.

4.Please use always a blank space between values and units. This is not coherent in your manuscript.

The error has been corrected.

5.In 2.1 is not clear the proportion of each component used in the mixing process. Could you clarify?

OK, the content of each component has been added.

6.In Table 1, please change the samples' reference to the 2nd line, and change also the name from "Number" to "Sample's Reference". Please do the same in Table 2;

Thank you. It has been corrected.

7.Please refer the real brand of "magics software" (line 98);

Specific brand of software has been added.

8.In line 99, I thing there is something wrong: "...and the printer is imported for printing." Please correct.

Thank you. It has been corrected.

9.In 2.5 the text is hard-to-read. Please rewrite it with the help of an English Native Speaker;

It has been reedited and a diagram has been added to illustrate it.

10.In 2.5 please include a diagram with the thermal cycle.

Already added a diagram.

11.Please format 2.6. All formulas need to be numbered;

OK, all formulas have been numbered.

12.Please explain why you have choosen Y≈ 2.1639;

Attach the formula to explain the calculation of Y.

13.Please include a Discussion in your paper, halping the reader to compare the results previously obtained in other similar works and yours;

Add references and compare with similar research results.

14.It would be expected a deeper discussion about the phenomena behind the results. Thus, please enrich your manuscript crossing experiences with other Researchers.

OK, references have been added to enrich the paper.

Finally, thank you for your careful correction. We have revised the paper according to your opinions.

Wish you a happy life.

Kind regards.

Round 2

Reviewer 1 Report

Manuscript ID: MDPI Materials-1642506_REV_2

Title: Enhancement Effect of Short Carbon Fiber on 3D Printed Alumina Ceramics (old title)

Strengthening Effect of Short Carbon Fiber on Mechanical Properties of 3D Printed Alumina Ceramics (new title)

The work still leave a lot of gaps though the experimental design was good. This may affect the readability. At least the authors should take some effort to underline the importance and the novelty of this short work (maybe improving the introduction?). The results are limited. In any case the editor can decide.

  1. The Title of the work is still not clear and needs to be making some sense. Please try to highlight the main contribution of the work, if any.

  1. I repeat my comment “In the SEM images of the samples presented: please modify the figure with only the scale and cutting the black bar with the magnification shown. The magnification doesn’t makes any sense”- please understand what the issue raised. It’s to remove the black bar below (showing the different parameters and show only the scale). This is a very basic thing. If you project this magnification on a big screen with this bar the magnification makes no meaning!

Hope you understood the comment.

Example: see in the attachment 

Author Response

Dear Reviewers,

Thank you very much for your valuable comments. This is very helpful for the improvement of our article.

  1. The Title of the work is still not clear and needs to be making some sense. Please try to highlight the main contribution of the work, if any.

In this regard, I redrafted the title "Strengthening Effect of Short Carbon Fiber Content and Length on Mechanical Properties of 3D Printed Alumina Ceramics". Because the main research focus of this paper is the effect of carbon fiber length and content on the toughening of ceramics, do you think this title is in line with the theme of this article? If you have other opinions or better ideas, please contact me for correction.

  1. I repeat my comment “In the SEM images of the samples presented: please modify the figure with only the scale and cutting the black bar with the magnification shown. The magnification doesn’t makes any sense”- please understand what the issue raised. It’s to remove the black bar below (showing the different parameters and show only the scale). This is a very basic thing. If you project this magnification on a big screen with this bar the magnification makes no meaning!

I'm so sorry, this is the first time I sent a paper so that I didn't realize this mistake. I have revised the SEM diagram in the correct way. I'm sorry I misunderstood your suggestion earlier.

Thank you for your patient guidance. We have revised the paper according to your opinions.

Wish you a happy life.

Kind regards

Reviewer 2 Report

Thanks for the revision. However, there are some errors such as year, authors name, name of the journal in the list of bibliography.

You should pay attention as it is an important part in publishing in international journals.

I'll notify them to the editor to consider it in the proof of the paper. A simple way is to add the DOI link of the paper with the references.

Best wishes

Author Response

Dear Reviewers,

OK, we have revised the references and added the DOI link. Thank you for your careful guidance. If you find any other problems, please feel free to contact us.

Wish you a happy life.

Kind regards.

Reviewer 3 Report

No comments.

Author Response

Dear Reviewers,

Thank you for your careful guidance to improve this paper. I hope I can cooperate with you in the future.

Wish you a happy life.

Kind regards.
